# TEPO: A Transferable EDA Prediction Method Based on Learngene Characterization

## Abstract

This paper introduces TEPO, a novel multi-task learning framework to optimize Electronic Design Automation (EDA) in integrated circuit (IC) design by addressing increasing complexity and the limitations of traditional independent design task approaches. TEPO systematically decomposes design knowledge into gene knowledge and class knowledge, referred to as Learngenes. The framework employs a dual-pathway architecture with an adaptive gating mechanism, enabling fine-grained control over knowledge activation and enhancing computational efficiency and interpretability. The VIT-GNN fusion processor integrates Vision Transformer (ViT) features from layout images with Graph Neural Network (GNN) features from circuit topology, spatially aligning them onto a unified 256×256 grid to preserve both global visual patterns and local structural relationships. Our approach tackles four critical challenges in EDA: knowledge fragmentation, feature integration, transferability, and data scarcity. The methodology involves pre-training an upstream model to extract Learngene, which initializes a downstream 12-layer Transformer model for various prediction tasks. Experiments are conducted on CircuitNet-N28, a dataset providing multi-modal features for Congestion, DRC violations, IR-drop, and a new thermal prediction task. TEPO demonstrates strong transferability, faster convergence, reduced data requirements, and lower computational costs while achieving superior performance.

## 1 Introduction

Electronic Design Automation (EDA) faces growing complexity as modern ICs require simultaneous optimization of Performance, Power, and Area (PPA) (Wang, 2016). Since PPA metrics are only available post-layout (Kawa et al., 2006; Lavagno et al., 2018), long iteration cycles hinder design efficiency. Early prediction of these metrics is critical for rapid defect identification, motivating the use of neural networks in EDA (Knechtel et al., 2020; Yu, 2023; Goswami & Bhatia, 2023). However, existing methods treat tasks in isolation, failing to exploit shared design patterns across stages (Shrestha & Savidis, 2024).

We propose **TEPO**, a transferable EDA optimization framework that decomposes knowledge into *gene knowledge* (universal IC design patterns) and *class knowledge* (task-specific expertise). This separation enables effective knowledge sharing and rapid adaptation to new tasks. TEPO features a dual-pathway architecture with matrix decomposition and adaptive gating: gene gates ($\sigma_{gene}$) regulate universal patterns, while class gates ($\sigma_{class} \odot g_{task}$) route task-specific knowledge, enhancing both efficiency and interpretability.

To integrate multi-modal inputs, we introduce a VIT-GNN fusion processor that combines layout features from Vision Transformers (ViT) with topology representations from Graph Neural Networks (GNNs). By aligning ViT patches and GNN nodes on a unified 256×256 spatial grid via nearest-neighbor mapping, our fusion preserves global visual structures and local connectivity.

Our key contributions are:

1. **Knowledge Decomposition for EDA**: We present the first framework to explicitly separate universal and task-specific knowledge in EDA via weight matrix decomposition $W_{\text{effective}} = W_{\text{gene}} + W_{\text{class}}$, addressing knowledge fragmentation.

2. **Multi-Modal Fusion**: The VIT-GNN processor bridges layout-image and circuit-graph modalities through spatial alignment, enabling comprehensive feature integration.

3. **Adaptive Knowledge Routing**: A dual-level gating mechanism allows fine-grained control over knowledge activation, improving flexibility and interpretability.

4. **Transferable Optimization**: TEPO achieves fast convergence and reduced data needs in both known and novel tasks by inheriting gene knowledge, demonstrating strong transferability under data scarcity.

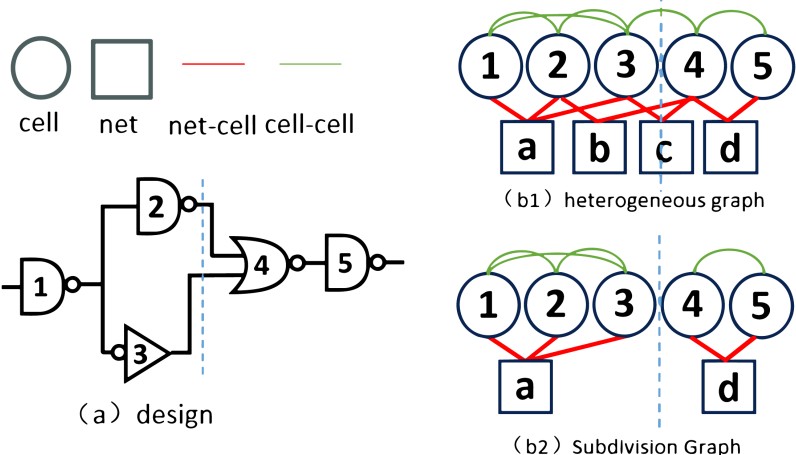

Figure 1: Heterogeneous graph construction and feature extraction for GNNs.

## 2 RELATED WORK

### 2.1 EARLY STAGE PREDICTION

For modern Electronic Design Automation (EDA), the design flow is characterized by its extensive chain and the difficulty of isolated point optimization (Dong et al., 2023). The key PPA metrics (Power, Performance, Area) that evaluate chip quality are only obtainable after the entire flow completes, leading to long iteration cycles. Therefore, Early Stage Prediction of final performance metrics is crucial for identifying potential defects early on (Ren & Hu, 2022; Kahng, 2022). This primarily includes:

- **Congestion Prediction**: This involves anticipating areas within the chip that are likely to experience routing resource shortages when numerous signal lines need to pass through a restricted area (Kirby et al., 2019). Prediction is based on netlist information, the placement of standard cells and macros, and the availability of routing resources.

- **DRC(Design Rule Check) Prediction**: This process aims to foresee which chip areas are most susceptible to violating manufacturing process design rules (Islam, 2022). It utilizes information such as component position, size, and orientation, the approximate routing paths and area occupancy of signal lines, and process design rules. Design rules themselves are a set of geometric and electrical constraints defined by the semiconductor manufacturer.

- **IR-drop Prediction**: This focuses on forecasting the voltage drop within the chip's power delivery network, which occurs due to current flowing through resistance (I*R). The prediction is made using the power network's topological layout and the power consumption information of individual module units (Xie et al., 2020). It's intrinsically linked to the chip's instantaneous current demands across various operating modes.

- **Thermal Prediction**: This involves predicting the regional temperature distribution of the chip during operation (Yan et al., 2025). The prediction takes into account the power consumption of various chip modules, material thermal characteristics, heat dissipation structures, and the ambient temperature in which the chip operates (Yu et al., 2025).

## 2.2 LEARNGENE

Learngene represents a paradigm shift in knowledge transfer, functioning as a "neural genetic code" (Bohacek & Mansuy, 2015) to compress and preserve pre-trained insights (Feng et al., 2023). By encapsulating critical common knowledge into modular fragments, it allows descendant networks to inherit essential information efficiently, avoiding the redundancy of traditional transfer methods (Feng et al., 2024).

This approach offers significant advantages over conventional fine-tuning. Its modular design enables targeted transfer with minimal computational overhead, while Learngene-initialized models demonstrate exceptional transferability. Experiments indicate convergence speeds up to 40% faster and robust generalization in low-data regimes, matching the performance of models trained with significantly more labeled data (Wang et al., 2022).

Despite its success, Learngene's application remains largely confined to computer vision and NLP. This contrasts with other mainstream techniques like Knowledge Integration and Diversion (KID) (Xie et al., 2024) and WAVE (Feng et al., 2025), which continue to focus on image processing tasks. Our work aims to bridge this gap by extending the Learngene framework to the challenging domain of Electronic Design Automation (EDA).

## 3 METHODOLOGY

Based on the four challenges (Knowledge Fragmentation, Feature Integration, Transferability and Scalability) in the EDA domain proposed above, we propose a novel method called TEPO to carry out the migration of EDA prediction tasks.

### 3.1 MULTIMODAL FUSION

For Graph Neural Network (GNN) models, they excel at capturing the topological features of a chip, including the relative positions and connectivity of nodes. However, GNNs often fall short in fully extracting the logical relationships of features and tend to overlook the underlying structure of the netlist (Ren et al., 2022; Ma et al., 2020).

In contrast, Vision Transformer (ViT) models are adept at capturing the geometric features of a chip, such as node positions, shapes, and orientations. Yet, ViTs lack node neighborhood information, making them less effective at capturing topological features.

To leverage the strengths of both, we simultaneously employ a GNN model to extract topological features and a ViT model to extract geometric features. The fusion of these two modalities yields a comprehensive fused feature that encompasses both geometric and topological information, leading to enhanced representational capabilities.

### 3.2 GNN FEATURE EXTRACTION

Our approach begins by constructing a heterogeneous graph directly from the netlist, incorporating both cell and net node types. To enhance training efficiency, we partition the entire graph into several subgraphs, a strategy inspired by Circuit GNN (Yang et al., 2022) in Figure 1. The feature set for cells, nets, and their interconnections is defined as follows:

- **Cell Features**: We capture the dimensions of each cell, represented by its width ($w$) and height ($h$), along with features derived from the grid location of its center point.

- **Net Features**: For each net, we record the total number of connected pins, and its maximum span in both the horizontal ($h$) and vertical ($v$) directions.

- **Cell-to-Cell Edge Features**: The connection between two cells is characterized by their Manhattan distance, reflecting their spatial proximity on the chip.
- **Net-to-Cell Edge Features**: Edges connecting a net to a cell are characterized by the precise $x/y$ coordinates of the pin that establishes the connection.

We will perform Linear Projection on the features of the graph, using a fully connected layer to project the feature vectors of different nodes onto a fixed-size dimension D. Subsequently, the projected results are input into the information between the HeteroGraphConv (Yang et al., 2022) fusion nodes, which includes:

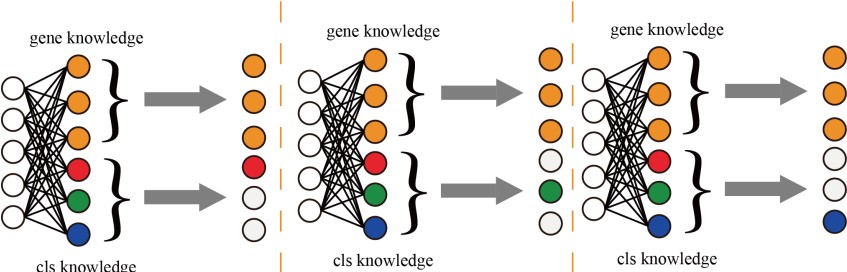

Figure 2: Learngene is applied to various types of downstream tasks.

1. **CFCNN**: This component is responsible for aggregating information from nets onto cells, effectively enriching cell representations with net-level context.

$$h_i^{(l+1)} = W_{\text{out}}^{(l)} \left( h_i^{(l)} + \sum_{j \in \mathcal{N}(i)} \left( W_{\text{in}}^{(l)} h_j^{(l)} \right) \odot \text{MLP}^{(l)}(e_{ij}) \right) \tag{1}$$

where $\mathbf{h}_i^{(l)}$ represents the cell representation at layer $l$, $\mathbf{W}_{\text{out}}^{(l)}$ and $\mathbf{W}_{\text{in}}^{(l)}$ are the output and input weight matrices at layer $l$ respectively, $\mathcal{N}(i)$ denotes the neighbors of cell $i$, $\mathbf{e}_{ij}$ is the edge feature from cell $i$ to cell $j$, $\text{MLP}^{(l)}$ is the Multi-Layer Perceptron at layer $l$, and $\odot$ denotes element-wise multiplication.

2. **SAGEConv**: To capture inter-cell relationships, SAGEConv enables information exchange directly between cells, allowing them to learn from their topological neighbors.

$$h_i^{(l+1)} = W^{(l)} \left( h_i^{(l)} \bigg\| \frac{1}{|\mathcal{N}(i)|} \sum_{j \in \mathcal{N}(i)} h_j^{(l)} \right) \tag{2}$$

where $\|$ denotes the concatenation operation.

3. **GraphConv**: Conversely, GraphConv aggregates cell-specific information onto nets, ensuring nets are informed by the characteristics of the cells they connect.

$$h_i^{(l+1)} = \sigma \left( \text{BN}^{(l)} \left( W_1^{(l)} \cdot h_i^{(l)} + W_2^{(l)} \cdot \sum_{j \in \mathcal{N}(i)} e_{ji} \cdot h_j^{(l)} \right) \right) \tag{3}$$

where $\sigma$ represents a nonlinear activation function. $\text{BN}^{(l)}$ stands for Batch Normalization at layer $l$. It normalizes the inputs of a layer by re-centering and rescaling them.

Finally, a Flatten by Position operation is applied. This critical step strategically places the learned features of each node onto their corresponding spatial locations within a unified Feature Map, preparing the data for subsequent processing.

### 3.3 ViT Feature Extraction

We utilize a $256 \times 256$ feature map to represent node information. This process begins with Patch Embedding, which involves dividing the input ViT grid map into fixed-size patches. Each resulting patch is then augmented with positional embeddings. These patches are subsequently linearly

projected into a fixed-dimensional space, forming a patch sequence. This patch sequence is then processed through a stack of multiple ViT Encoder layers to extract geometric features.

The GNN's feature space is projected onto the ViT's 256×256 feature map. To align the ViT's patch features with the GNN's node features, a feature difference loss is employed, which supervises the model's learning for the prediction task. These integrated features then serve as input for the downstream model.

## 3.4 GENERATION AND EXTRACTION OF LEARNGENE

In the TEPO framework, we introduce a novel knowledge gene extraction mechanism that decomposes the traditional weight matrix $W \in \mathbb{R}^{d_{out} \times d_{in}}$ into two semantically meaningful components: gene knowledge and class knowledge. This decomposition is based on Singular Value Decomposition (SVD), a technique that factors a matrix into three components, $W = U\Sigma V^T$, where $U$ and $V$ are orthogonal matrices and $\Sigma$ is a diagonal matrix containing the singular values in descending order.

We leverage the property of singular values to separate universal and task-specific knowledge. The singular values in $\Sigma$ represent the importance of each component. By partitioning these values, we can obtain two low-rank approximations of the original weight matrix.

$$W_{gene}^{(l)} = U_{gene}\Sigma_{gene}V_{gene}^T \quad \text{and} \quad W_{class}^{(l)} = U_{class}\Sigma_{class}V_{class}^T \tag{4}$$

where $l$ is the layer index, and the matrices are defined as follows:

- $U_{gene} \in \mathbb{R}^{d_{out} \times k_{gene}}$, $\Sigma_{gene} \in \mathbb{R}^{k_{gene} \times k_{gene}}$, and $V_{gene}^T \in \mathbb{R}^{k_{gene} \times d_{in}}$. These matrices are derived from the top $k_{gene}$ singular values and their corresponding singular vectors. $W_{gene}^{(l)}$ represents the **gene knowledge**, which captures the most dominant and universal features of IC design.

- $U_{class} \in \mathbb{R}^{d_{out} \times k_{class}}$, $\Sigma_{class} \in \mathbb{R}^{k_{class} \times k_{class}}$, and $V_{class}^T \in \mathbb{R}^{k_{class} \times d_{in}}$. These are derived from the remaining $k_{class}$ singular values and vectors. $W_{class}^{(l)}$ represents the class knowledge, which captures the less dominant but task-specific information.

The effective weight matrix for a given layer is then the sum of these two components, as expressed in Equation (3).

$$W_{effective}^{(l)} = W_{gene}^{(l)} + W_{class}^{(l)} \tag{5}$$

This approach ensures that the decomposition is not arbitrary but is based on the inherent structure of the weight matrix, thereby addressing the identifiability concern raised by the reviewer.

The diagonal elements of the matrix $\Sigma$, known as singular values, are ordered in descending magnitude. These singular values mathematically represent the most significant to least significant information components within the matrix (Strang, 2012). Our approach leverages this property of SVD to ensure the uniqueness and semantic meaning of the decomposition. Specifically, we first perform an SVD on the original 768x768-dimensional weight matrix W. We then partition the knowledge based on the singular values in $\Sigma$:

Gene Knowledge: We construct the matrix by extracting the top 512 largest singular values from the diagonal matrix $\Sigma$ and multiplying them with their corresponding U and V matrices. Since these largest singular values represent the most dominant and general information within the matrix W, we define them as the "gene knowledge" shared across all tasks.

Class Knowledge: Correspondingly, we use the remaining 256 singular values to construct the matrix. These smaller singular values represent the less dominant, more specific information, which we define as the task-specific "class knowledge."

By strictly partitioning based on the magnitude of the singular values, we assign a clear semantic role to both $W_{gene}^{(l)}$ and $W_{class}^{(l)}$: $W_{gene}^{(l)}$ captures general patterns, while $W_{class}^{(l)}$ captures specific details.

## 3.5 INHERITANCE OF LEARNGENE

In the preceding step, we have successfully saved the pre-trained weight parameters for both gene knowledge and class knowledge models. These pre-trained weights serve as the cornerstone for subsequent model adaptation and task-specific fine-tuning, encapsulating a wealth of learned patterns and features from the initial training phase. When confronted with downstream tasks, a strategic approach is adopted for model initialization. Gene knowledge embodies fundamental biological and genetic principles that are often transferable across various related tasks. By leveraging pre-trained gene knowledge, the model can start from a more informed state, reducing the amount of data and computational resources required for convergence during the training on downstream tasks.

In the scenario where existing task types are involved, based on the specific nature of the downstream task type, the corresponding class knowledge is carefully selected. Different task types rely on distinct aspects of class knowledge, which capture task-specific semantic and structural information. For tasks that fall within different categories, a sophisticated gating mechanism is employed. This gating mechanism selectively activates the relevant class knowledge weight parameters while randomly initializing the remaining parts of the class knowledge. Through this mechanism, the model can focus on the most pertinent knowledge for the given task, enhancing its efficiency and performance. Let the input patch sequence be $X \in \mathbb{R}^{B \times N \times d_{\text{in}}}$, where $B$ is the batch size, $N$ the sequence length, and $d_{\text{in}}$ the input feature dimension. The KIND linear layer decomposes transformation into a shared ("gene") path and a task-specific ("class") path.

**Key learnable components.** We define three core quantities:

- The **shared scaling vector**:
$$\sigma_{\text{gene}} \in \mathbb{R}^{d_g},$$
which applies element-wise rescaling to the shared low-rank subspace.

- The **task-specific scaling vector**:
$$\sigma_{\text{class}} \in \mathbb{R}^{d_c},$$
which modulates importance across the task-adaptive subspace.

- The **hard task gating vector** for task $\tau$:

$$g_i^{(\tau)} = \begin{cases} 1, & \text{if } i \in [s_\tau, e_\tau), \\ 0, & \text{otherwise,} \end{cases} \quad \text{so that } g^{(\tau)} \in \{0,1\}^{d_c}.$$

This enforces sparse activation of only the dimensions allocated to task $\tau$.

Given projection matrices $U_g \in \mathbb{R}^{d_{\text{in}} \times d_g}$, $V_g \in \mathbb{R}^{d_g \times d_{\text{out}}}$, $U_c \in \mathbb{R}^{d_{\text{in}} \times d_c}$, and $V_c \in \mathbb{R}^{d_c \times d_{\text{out}}}$, the output of the KIND linear layer is:

$$Y = \underbrace{(XU_g \odot \sigma_{\text{gene}})V_g}_{\text{shared path}} + \underbrace{(XU_c \odot \sigma_{\text{class}} \odot g^{(\tau)})V_c}_{\text{gated task-specific path}} + b, \tag{6}$$

where $\odot$ denotes broadcasting element-wise multiplication over the $(B,N)$ dimensions, and $b \in \mathbb{R}^{d_{\text{out}}}$ is an optional bias.

**Ablation: gating vs. additive mixing.** To evaluate the necessity of explicit task routing, we compare against an *additive mixing* variant that removes the gating mask:

$$Y^{\text{add}} = (XU_g \odot \sigma_{\text{gene}})V_g + (XU_c \odot \sigma_{\text{class}})V_c + b. \tag{7}$$

This baseline allows all tasks to access the full class subspace, potentially causing cross-task interference. Our ablation study measures the performance gap between $Y$ (gating) and $Y^{\text{add}}$ (additive) across heterogeneous multi-task settings.

Conversely, when a new task type is encountered, one that has not been previously seen during the training or knowledge extraction phases, only gene knowledge is used to initialize the model's weight parameters. Because gene knowledge provides a broad and general foundation that can potentially adapt to novel tasks. Since no relevant class knowledge exists for the new task type, none

of the class knowledge components will be used. Instead, all dimensions of the class knowledge will be initialized randomly. This random initialization allows the model to explore and learn the unique features and requirements of the new task from scratch, while still benefiting from the initial guidance provided by the gene knowledge. This approach strikes a balance between leveraging existing knowledge and being flexible enough to accommodate unforeseen task variations, enabling the model to exhibit robust performance across a wide spectrum of downstream tasks.

# 4 EXPERIMENTS

## 4.1 DATASETS

We conduct experiments on CircuitNet-N28 (Chai et al., 2022; Xun et al., 2024), a dataset that provides multi-modal features (image and graph) to support four cross-stage prediction tasks in back-end design: Congestion prediction, DRC (Design Rule Check) violations prediction, and IR-drop prediction. N28 refers to the 28nm planar technology. The dataset represents IC features in graph format, including: Macro Region, Cell density, RUDY (Routing Utilization and Density), Pin configuration, Congestion, DRC violations, Instance power, Signal arrival timing window and IR-drop. The thermal prediction labels are generated through thermal simulation using hotspot, which can simulate the thermal labels of the chips based on the existing LEF/DEF files.

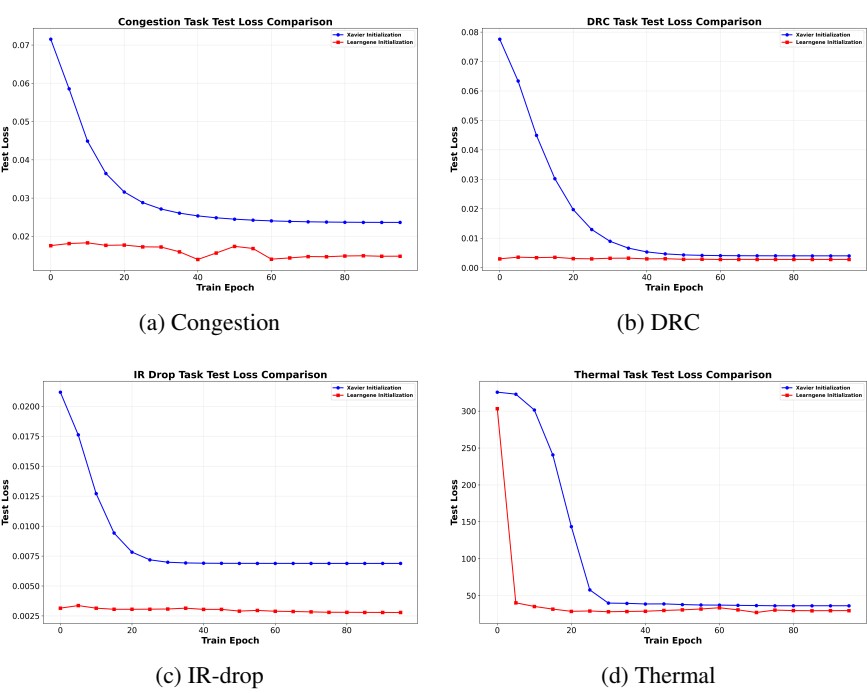

(a) Congestion          (b) DRC

(c) IR-drop          (d) Thermal

Figure 3: Visualization of convergence speed of TEPO and Xavier methods on downstream tasks. From the changing trend of the figure, it can be intuitively seen that TEPO has a fast convergence speed and extremely strong transferability.

## 4.2 BASIC SETTING

First, we extract 100 samples from the CircuitNet-N28 raw data to form the training set and 20 different samples to form the testing set. It should be emphasized that each sample here represents a complete IC design, and the volume of data is extremely large. Each sample contains tens of thousands of logical unit information, as well as multimodal topological and physical layout data. These lists are then fed as input to both the Vision Transformer (ViT) and Graph Neural Network (GNN) models. We set the fusion coefficient to 0.5, meaning we take half of the output from each of the ViT and GNN models for feature fusion, resulting in a 256×256 grid as the fused feature.

For the pre-training model, we employ a 12-layer Transformer model, with the fused features serving as its input. The pre-training objectives are the congestion prediction, DRC prediction, and IR-drop prediction tasks. The sum of the losses from these three tasks is used for backpropagation.

The downstream 12-layer Transformer task model also takes the fused features as input but operates in a single-task training mode. Specifically, it's trained for individual tasks, including: same-category tasks for Congestion, same-category tasks for DRC, same-category tasks for IR-drop, and a new task type, thermal prediction.

The Key training details have been explained in the supplementary materials and can be found in the README file.

### 4.3 Evaluation and Results

We evaluate TEPO in downstream tasks using a pre-trained Learngene for model initialization, comparing against random (Xavier) initialization and state-of-the-art EDA models.

**Transfer Learning Setup:** We assess performance on three established tasks—congestion, DRC, and IR-drop prediction—and one novel task, thermal prediction, to test generalization beyond pre-training categories. For known tasks, both gene and class knowledge are used (with task-specific gating); for thermal prediction, only gene knowledge is activated, as no corresponding class knowledge exists.

| Method | Congestion↓ | DRC↓ | IR-drop↓ | Thermal↓ |
|--------|-------------|-------|----------|----------|
| Xavier | 0.023 | 0.004 | 0.068 | 36.214 °C |
| TEPO | **0.014** | **0.002** | **0.027** | 29.933 °C |

Table 1: TEPO uses Learngene for weight initialization, whereas Xavier adopts random weight initialization. This is reflected in their final performance on downstream tasks, with the evaluation metrics being Mean Squared Error (MSE) and Celsius degrees.

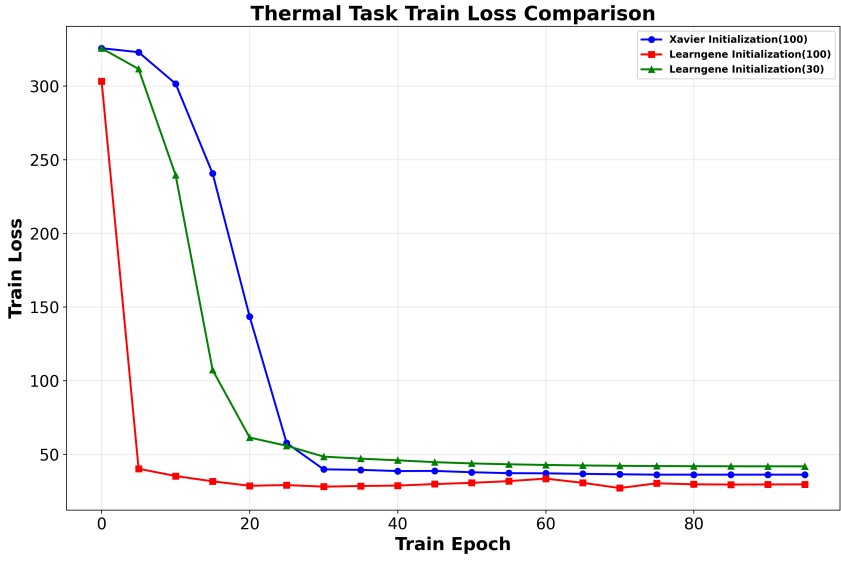

Figure 4: For the Thermal task, Learngene uses 100 and 30 training data respectively, while Random uses 100 training data.

**Convergence and Performance:** As shown in Figure 3 and Table 1, TEPO achieves significantly faster convergence than Xavier initialization across all tasks. On congestion, DRC, and IR-drop, TEPO reaches near-optimal performance within few epochs, while Xavier requires substantially more iterations. For the unseen thermal task, TEPO converges approximately 75% faster despite

lacking task-specific class knowledge, demonstrating strong transferability through learned universal design patterns.

**Data Efficiency:** Figure 4 and Table 2 show that TEPO maintains high accuracy even under limited training data (e.g., 50% or 30% of full set), outperforming Xavier by large margins, especially in low-data regimes.

**Comparison with SOTA Models:** In end-to-end performance (Table 3), TEPO surpasses existing EDA models including CircuitNet (GPDL, RouteNet), NetlistGNN, GCN, GAT, and SAGE, validating the effectiveness of Learngene-based initialization and multi-modal fusion.

In addition, we present the specific data of the trained models in Table 1, showing that TEPO also achieved performance improvements: the accuracy of the Congestion prediction task was improved by approximately 39.13%, DRC by about 50%, IR-drop by around 60.29%, and Thermal by roughly 17%.

In Figure 4, we trained the model using varying amounts of training data for new task type (Thermal prediction). Notably, with merely 30 training data points, the TEPO method achieved predictive accuracy comparable to that of the Xavier method trained with 100 data points. This demonstrates that, due to the substantial common knowledge embedded within its Learngene, the TEPO method can reduce the required training data by approximately 70% to achieve the same level of accuracy, leading to significant savings in computational overhead. This is particularly valuable for the EDA field, where labeled data is extremely scarce. And in Table 2, the specific final performance of the models can be observed.

| Method | Training Data Size | Thermal↓ |
|--------|--------------------|----------|
| Xavier | 100 | 36.214 °C |
| TEPO | 30 | 41.857 °C |
| TEPO | 100 | 29.933 °C |

Table 2: The model performance achieved by TEPO and Xavier when training models with different amounts of training data. The temperature under thermal represents the difference in temperature with the label.

| Method | Congestion↓ | DRC↓ | IR-drop↓ | Thermal↓ |
|--------|-------------|------|----------|----------|
| GCN | 0.329 | 0.075 | 0.047 | 44.373 °C |
| RouteNet | 0.021 | 0.005 | **0.014** | 30.154 °C |
| NetlistGNN | 0.097 | 0.073 | 0.046 | 41.988 °C |
| SAGE | 0.327 | 0.074 | 0.047 | 44.373 °C |
| GPDL | 0.022 | 0.006 | 0.025 | 38.471 °C |
| GAT | 0.325 | 0.075 | 0.047 | 44.373 °C |
| TEPO | 0.014 | **0.002** | 0.027 | 29.933 °C |

Table 3: The performance of various models and TEPO was evaluated across four tasks: Congestion, DRC, and IR-drop, and Thermal. Prediction accuracy for Congestion, DRC, and IR-drop was measured using Mean Squared Error (MSE). For the Thermal task, prediction accuracy was quantified by the error in predicted Celsius degrees.

To demonstrate our model's efficacy, we conducted a comparative analysis of TEPO against the established neural network model within our dataset, as well as several other models frequently utilized in the Electronic Design Automation (EDA) domain, as shown in Table 1. Due to the use of learning genes, TEPO not only outperforms conventional models in convergence speed, but also, because of the application of fusion features, TEPO can extract data features more accurately and has excellent model performance. For the three previously encountered task categories: Congestion, DRC, and IR-drop, TEPO consistently achieved strong performance, proving to be fully competitive with traditional predictive models. Furthermore, TEPO exhibited remarkable transferability on the novel task type, Thermal. Its capacity to assimilate a significant body of common knowledge pertinent to IC (Integrated Circuit) design enabled it to deliver highly favorable results on this new

challenge. In contrast, other models demonstrated inherent limitations in their ability to rapidly adapt to unfamiliar tasks.

## 4.4 ABLATION

To evaluate the contribution of each component in our hybrid architecture, we conduct an ablation study by comparing variants of the full model—ViT+GNN—against its individual counterparts. Specifically, we compare ViT+GNN with standalone ViT under a ViT-based backbone, and with standalone GNN under a GNN-based backbone. As shown in Table 4, when integrated with GNN, the ViT backbone achieves significantly lower IR-drop and total loss (0.03766 vs. 0.09448 for IR-drop; 0.07797 vs. 0.13304 for total loss), demonstrating that the GNN module effectively enhances physical constraint modeling. This result validate that the synergy between ViT and GNN consistently improves prediction accuracy.

| Backbone | Method | Congestion | DRC | IR-drop | Total Loss |
|----------|--------|-----------|-----|---------|-----------|
| ViT | ViT+GNN | 0.03376 | 0.00655 | 0.03766 | **0.07797** |
|  | ViT | 0.03218 | 0.00638 | 0.09448 | 0.13304 |
| GNN | ViT+GNN | 0.03448 | 0.00827 | 0.01660 | **0.05934** |
|  | GNN | 0.03492 | 0.00698 | 0.01922 | 0.06111 |

Table 4: In the input stage of the model, the fusion features of ViT and GNN are used, or the features of either Vit or GNN are used separately for comparison on the Vit and GNN networks

| Backbone | Method | Congestion | DRC | IR-drop | Total Loss |
|----------|--------|-----------|-----|---------|-----------|
| ViT | TEPO | 0.03027 | 0.00256 | 0.03219 | **0.06502** |
|  | MLP | 0.03376 | 0.00655 | 0.03766 | 0.07797 |
| GNN | TEPO | 0.03441 | 0.00705 | 0.01608 | **0.05754** |
|  | MLP | 0.03448 | 0.00827 | 0.01660 | 0.05934 |

Table 5: Both use fusion features as input. One employs MLP, while TEPO uses Learngene architecture

Furthermore, on the basis of retaining the fusion features, we used the knowledge diversion architecture of TEPO for prediction. In contrast, we directly used MLP and connected three output heads for prediction. As shown in Table 5, the difference in the prediction results of our architecture is significantly smaller.

## 5 CONCLUSION

This paper proposes TEPO, an innovative multi-task learning framework, aimed at addressing the increasing complexity of Electronic Design Automation (EDA) in IC (integrated circuit) design. By introducing multimodal feature fusion and Learngene, TEPO has significantly enhanced the efficiency and accuracy of EDA prediction tasks.

The experimental results show that TEPO performs well in existing task categories, and demonstrates outstanding transferability and superior performance on new tasks such as Thermal prediction. Compared with traditional methods, TEPO not only achieves a faster convergence speed but also significantly reduces the amount of training data required to achieve the same performance level, thereby significantly saving computational costs.

The proposal of TEPO has opened up new research directions for AI-driven EDA tools, especially in efficient learning by leveraging common design patterns and task-specific knowledge. Future work can explore the extension of TEPO to a wider range of EDA applications, such as physical design optimization or design space exploration, and further study its generalization ability at different process nodes and more complex chip architectures.

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
