# OpenReview forum: "TEPO: A Transferable EDA Prediction Optimization Method Based on Learngene Characterization"
_ICLR.cc/2026/Conference — Submitted to ICLR 2026_

### Official Review · Reviewer_1vTs · 2025-10-27

**Soundness:** 2
**Presentation:** 2
**Contribution:** 2
**Rating:** 4
**Confidence:** 4

**Summary:**

The paper proposes TEPO, a transferable framework for EDA prediction. TEPO factorizes each layer’s weights into task-agnostic “gene knowledge” and task-specific “class knowledge” (via an SVD-style decomposition) and combines them through dual-path gating during transfer. On the feature side, layout images (ViT) and netlist/topology features (GNN) are spatially aligned and fed to a Transformer. Experiments cover standard backend targets (congestion/DRC/IR-drop) and include an additional Thermal task aimed at testing cross-category transfer. The manuscript reports faster convergence and lower final errors than Xavier initialization and several EDA baselines.

**Strengths:**

- Clear modular design. The weight factorization plus gating is conceptually coherent; the multi-modal alignment (layout + netlist) is a sensible engineering pipeline.
- Transfer signal. Results indicate faster convergence and improved endpoints over vanilla initialization across multiple tasks, suggesting value as a pretraining/initialization strategy.
- Practical motivation. Addressing multi-task transfer in data-constrained EDA settings is relevant to downstream design flows.

**Weaknesses:**

- Missing efficiency and reproducibility details. Training time, inference runtime/latency, computing resources (hardware, memory), and model size/compute (parameters/FLOPs) are not reported, limiting assessment of the claimed efficiency and hindering reproduction.
- Limited statistical reporting. The manuscript does not provide evaluations over multiple random seeds or dispersion measures (e.g., mean ± std, confidence intervals) and does not report significance testing, weakening the strength of the empirical claims.
- Thermal label transparency. The Thermal ground truth appears to be taken directly from the referenced dataset (so transparency would largely inherit from the dataset), yet the paper does not explain how those labels are defined or verified within the dataset context; a short clarification would help readers assess fidelity and reproducibility.
- Task coverage gap (timing). Timing prediction (e.g., slack/WNS/TNS) is a key objective in EDA because it directly guides placement optimization and impacts final performance and PPA quality. Recent timing-prediction frameworks such as E2ESlack [1] and PreRoutGNN [2] explicitly address pre-routing timing prediction using global pretraining and local delay learning. Moreover, cross-stage optimization work such as LaMPlace [3] highlights that improving timing correlation during placement leads to better downstream metrics, underscoring the importance of evaluating timing transfer.

**Questions:**

1) Gene/Class ranks and gating. How are the per-layer ranks for “gene” vs. “class” chosen (fixed ratios or data-driven), at what granularity is gating applied (per-layer, per-head, or per-channel), and what sensitivity is observed on convergence, final accuracy, and compute/latency?
2) Thermal ground truth. If Thermal labels are inherited from the dataset, how are those labels defined (solver, boundary conditions, grid/resolution, dataset QA) and how is leakage between training and test designs or stages avoided?
3) Timing prediction. Since timing is one of the most critical tasks in EDA and directly guides placement for better downstream performance, why is timing not included among the evaluated tasks, and how would TEPO be expected to transfer to timing (e.g., pre-routing slack/WNS/TNS) compared with baselines such as E2ESlack [1], PreRoutGNN [2]?

[1] Bodhe, S., Zhang, Z., Hamidizadeh, A., Kai, S., Zhang, Y., & Yuan, M. (2025). E2ESlack: An End-to-End Graph-Based Framework for Pre-Routing Slack Prediction. arXiv preprint arXiv:2501.07564.

[2] Zhong, R., Ye, J., Tang, Z., Kai, S., Yuan, M., Hao, J., & Yan, J. (2024, March). Preroutgnn for timing prediction with order preserving partition: Global circuit pre-training, local delay learning and attentional cell modeling. In Proceedings of the AAAI Conference on Artificial Intelligence (Vol. 38, No. 15, pp. 17087-17095).

[3] Geng, Z., Wang, J., Liu, Z., Xu, S., Tang, Z., Kai, S., ... & Wu, F. LaMPlace: Learning to optimize cross-stage metrics in macro placement. In The Thirteenth International Conference on Learning Representations.

---

> ### Author Response · Authors · 2025-11-27
> **In response to your question**
>
> **W1**
> Thank you for pointing out this oversight we made. We have added this content to the experimental section of the article and provided a detailed README in the supplementary materials to help you and others reproduce our article.
>
> **W3**
> The labels for thermal prediction were generated through thermal simulation using hotspot, which can simulate the thermal labels of chips based on existing LEF/DEF files and has been applied in multiple references [1]. We have also added explanations in the 4.1 DATASETS section of the article.
>
> **W4**
> The issue you pointed out is extremely crucial. Since the label of the time series information is stored on the edge rather than the point we are currently targeting, we did not initially treat it as a task. We are conducting the related work of Timing prediction. Our current approach is to connect two points into an edge and determine the labels of these two points by pairwise pairing. Specifically, there are four delay information between the two points. the net delay in early and rise corner. the net delay in late and rise corner. the net delay in early and fall corner and the net delay in late and fall corner.  In this way, Timing prediction can also be regarded as a new task similar to heat prediction and be included in our experiment.
>
> **Q1**
> Regarding Question 1, we followed an existing setting. We used 512 and 256 dimensions mainly based on the specific settings in Reference [1]. This number is not uniquely determined and can be changed. However, the change in dimension will not have a significant impact on performance, as long as the gene dimension is ensured to be greater than the class dimension.
> The specific implementation of the gating mechanism is very simple and utilizes the principle of matrix multiplication. We have a weight matrix, and the gate acts as a column vector. Multiplying the weight matrix by the column vector, the result of the matrix multiplication is 0 when the corresponding matrix element is 0. When the corresponding matrix element is 1, the multiplication result is the original value. This allows us to filter out the parts we want to use: set the unused parts to 0 and the needed parts to 1. We have also added the exact gating equation in the article 3.5 INHERITANCE OF LEARNGENE .
> The final convergence judgment is that if the loss of the MSE we adopt shows a continuous increase or no longer decreases, we consider it to have converged.
>
> **Q2**
> We have already partially answered this question in W3. For each sample file, the chip data it contains is different, and it itself has the labels of congestion, drc, and irdrop tasks. We only need to use different samples for the training set and the test set, and there will be no overlap. The heat prediction is based on the LEF/DEF original file of the sample file using the label generated by hotspot. The results we generate correspond one-to-one according to the file names of the samples, and there will be no overlap.
>
> **Q3**
> Thank you for reiterating this issue here. We also recognize that this is indeed a very crucial problem worth your repeated mention. We have provided relevant answers in the above-mentioned W4, and for the CircuitNet dataset we use, other tasks do not rely on time series information for guidance.
>
> **References**
> [1] Xie, Yucheng, et al. "Kind: Knowledge integration and diversion for training decomposable models." **ICML**2025

---

### Official Review · Reviewer_C7yg · 2025-10-30

**Soundness:** 2
**Presentation:** 2
**Contribution:** 2
**Rating:** 4
**Confidence:** 4

**Summary:**

The paper proposes TEPO, a transferable multi-task framework for EDA prediction. TEPO fuses ViT features from layout images with GNN features from circuit topology by spatially aligning both on a 256×256 grid, and  decomposes model weights into “gene” (shared) vs “class” (task-specific) components with adaptive gating to route knowledge per task. The system is pre-trained on congestion, DRC, and IR-drop and then transferred to downstream tasks, including a new thermal prediction task. Experiments on CircuitNet-N28 claim faster convergence, better data efficiency, and improved accuracy over random (Xavier) initialization and several SOTA baselines.

**Strengths:**

1. There is Clear motivation for early, transferable prediction across EDA tasks.
2. The article has few writing errors and the charts are clear in meaning.

**Weaknesses:**

1. TEPO underperforms RouteNet on IR-drop in Table 3 (0.027 vs 0.014), so “surpasses existing EDA models” needs qualification by task.
2. CircuitNet-N28 provides a large amount of data on different designs. Only 100 designs were used for training and 20 designs for testing, which raises concerns about the effectiveness and scalability.
3. The selected SOTA is not the current or recent single-task SOTA.
4. This article lacks ablation experiments to prove the effectiveness of each part of the design.
5. This multi-task learning method should have a large number of gradient conflicts among different tasks, but no solution to the gradient conflicts has been seen. So, I am skeptical that the experimental results are better than those of models specifically designed for a single task.

**Questions:**

1.CircuitNet-N28, this dataset has no thermal ground truths. How did you obtain the labels for the experiment? Any validation vs. physics-based solvers?
2.What is the exact feature difference loss between ViT patches and GNN nodes? Any alternatives tested (e.g., cross-attention alignment)?
3.Table 2 lacks 30-sample results, is not matched with the description before,
4. Please provide the mathematical form of σ_gene, σ_class, and g_task, and an ablation bewteen gating and simple additive mixing.

---

> ### Author Response · Authors · 2025-11-27
> **In response to your question**
>
> **W1**
> Yes, your observation is very meticulous. As you said, this is a multi-task learning method. We do not focus on a single task, so it is inevitable that there will be performance differences in a single task. However, we perform better in all other tasks.
>
> **W2**
> Regarding this issue, I have provided an explanation in section 4.2 BASIC SETTING. You may not have noticed that each sample includes tens of thousands of chip data designs, not just 100 designs; it is 100 samples.
>
> **W3**
> The issue you pointed out is quite sharp. I have attempted to supplement this part of the experiment. Following the suggestions of another reviewer, I compared it with NetTAG, DeepGate4, and FlowTuner, and it was successful in operation. However, since their model architectures are all mixed predictions of nodes and subgraphs, we are completely node-level predictions. We can't make comparisons.
>
> **W4**
> We have supplemented the ABLATION experiments in the fusion feature section of 4.4 ABLATION, including the feature fusion comparison of Vit and GNN between single GNN and single Vit features. I believe that the comparison with Xavier in the learngene section has already been able to prove its effectiveness. The initialization using learngene has indeed been improved compared to the method not using them. As for the comparison of the latest algorithms, it has been supplemented and mentioned in the previous question.
>
> **W5**
> Just as you said, there are indeed conflicts among multiple tasks. Our solution is that learngene is divided into gene knowledge and cls knowledge. gene knowledge contains the common parts among multiple tasks. The cls knowledge section tries to retain their differences as much as possible. As for comparison with models specifically designed for single-task operations, I believe your consideration is extremely correct. The advantage of our model lies in migration, which saves the time and data volume required for training downstream tasks.
>
> **Q1**
> The labels for thermal prediction were generated through thermal simulation using hotspot, which can simulate the thermal labels of chips based on existing LEF/DEF files and has been applied in multiple references [1]. We have also added explanations in the 4.1 DATASETS section of the article.
>
> **Q2**
> For this issue, the number of nodes in Vit is less than that in GNN. All its nodes are distributed on a 256*256 grid, while the number of GNN nodes is the number of cells. A Vit grid point often contains multiple GNN cells. Our approach is to align the features of the grid points of a Vit with the features of multiple cells of its corresponding GNN. This will have two effects: one is to make the features of the GNN similar to those of the Vit, and the other is to make the features of multiple cells of the GNN similar. The specific operation steps are divided into two steps. First, take the average of the features of all cells at this point on the GNN to calculate the MSE and make its loss as low as possible. The second step is to calculate the MSE by using the features of the grid points in the Vit and the average features obtained in the previous step to reduce its loss. The advantage of doing so is that points that are physically close in space can express similar features, which is in line with reality and can also express logical features. Many works are based on this point, such as reference [2].
>
> **Q3**
> Thank you for your careful reading. I'm very ashamed that this is a writing mistake that I didn't notice during the writing and review process. It has been corrected in the article now.
>
> **Q4**
> We are very sorry that due to the length of the article, we did not elaborate in detail in this part. We have added the content you requested in the 3.5 INHERITANCE OF LEARNGENE section of the article.
>
> **References**
> [1] Shi, Yunqi, et al. "Open3DBench: Open-Source Benchmark for 3D-IC Backend Implementation and PPA Evaluation."
> [2] Yang, Shuwen, et al. "Versatile multi-stage graph neural network for circuit representation." **NIPS**2022

---

### Official Review · Reviewer_RtUC · 2025-10-30

**Soundness:** 3
**Presentation:** 1
**Contribution:** 2
**Rating:** 0
**Confidence:** 5

**Summary:**

This work proposed a transferable EDA prediction optimization method by Learngene, which is a weights initialization method by leveraging singular value decomposition (SVD) on the pre-trained weights. By this initialization method, this work performs better than random (Xavier) initialization. Besides, this work used VIT and GNN for multi-modal fusion to improve quality.

**Strengths:**

1. This work first leveraged Learngene in the EDA prediction task.
2. The performance of this work is better than random initialization.

**Weaknesses:**

1. The experiments of this work are too weak to verify the transferability of their method. As the title of this paper suggests, this work targets the transferable learning problem in the EDA domain. However, the authors only compare their method with random (Xavier) initialization.
1. The paper lacks a detailed explanation of data distribution, especially since the authors only select 120 samples (100 for training, 20 for testing) from a large-scale CircuitNet-N28 dataset (over 10K samples). Meanwhile, I think experimenting on the whole CircuitNet dataset will be more convincing.
1. The experiment settings are wrong for the thermal prediction task. As shown in Tables 1/2/3, I can not understand why the value of temperature is used to evaluate the model performance. In my opinion, the authors should use MSE, which is used for Congestion, DRC, and IR-drop predictions. Meanwhile, the authors didn't mention how to generate the thermal data in the experimental section.

**Questions:**

1. The title of this paper consists of "prediction" and "optimization". I think the paper only propose prediction method without optimization?
2. The pre-training loss contains the losses of downstream tasks. Therefore, I think is a multi-stage training strategy rather than transfer learning.

---

> ### Author Response · Authors · 2025-11-27
> **In response to your question**
>
> **W1**
> Oh, regarding this issue, at the end of the 4.3 EVALUATION AND RESULTS section in the text, we provided a considerable number of existing algorithms for result comparison. The previous Xavier initialization was merely our attempt to demonstrate the effectiveness of our method, with the key point being that the method is practical and feasible.
>
> **W2**
> I explained in the BASIC SETTING section of 4.2 that each sample contains tens of thousands of chip data designs. It's not just 100 designs, but 100 samples. I think if you really read it carefully, you won't ignore this information. As for conducting experiments on the entire CircuitNet dataset, oh my goodness, I think at least a 9090gpu is needed to complete this experiment, after all, we only use a 3090gpu.
>
> **W3**
> The thermal prediction labels are generated through thermal simulation using hotspot, which can simulate the thermal labels of the chips based on the existing LEF/DEF files. The temperature value here is the difference between the temperature predicted by the model and that of the thermal label, and its meaning can be regarded as equivalent to that of MSE. To take care of you and help you understand effectively, we have provided more detailed explanations in the article.
>
> **Q1**
> Oh my goodness! I'm very sorry. I really didn't expect that you would consider this a key question. I merely wanted to express an optimization of the prediction method, that is, to adopt another effective prediction method. I can't help but feel deeply guilty here. Out of respect for your opinion, we have changed the title. We hope you will give us an extra 5 points for the fact that we have solved this important problem. After all, you have two problems in total, and we have perfectly solved one.
>
> **Q2**
> I'm sorry that I can't agree with your opinion on this issue. We have not included any pre-training losses in the downstream tasks at all. The training for the downstream tasks uses a new model without any training. We only used learngene for initialization when initializing the model weights. If you do not agree with this point, please feel free to reply. If you think our explanation has successfully answered your doubts, please give us another 5 points to support our work.

---

### Official Review · Reviewer_RfUN · 2025-11-01

**Soundness:** 2
**Presentation:** 2
**Contribution:** 2
**Rating:** 4
**Confidence:** 3

**Summary:**

The paper proposes TEPO, a transfer learning framework for chip design prediction tasks. It combines a GNN (circuit topology) and ViT (layout image) by aligning features on a shared 256×256 grid. It then decomposes model weights via SVD into “gene” (universal) and “class” (task-specific) components (“Learngene”). These components are used to initialize a downstream transformer for new tasks. TEPO is evaluated on CircuitNet-N28 for congestion, DRC, IR-drop, and a new thermal task, and claims better accuracy, faster convergence, and better data efficiency than Xavier initialization and several existing EDA models.

**Strengths:**

* The SVD-based split of weights into transferable “gene” vs task-specific “class” knowledge is an interesting, explicit formulation of reusable IC design knowledge.
* The multimodal fusion of graph (netlist/topology) and layout image features is reasonable for EDA and is implemented end-to-end.
* TEPO shows faster convergence and better sample efficiency, including on a task (thermal) not seen during pretraining.
* Tables and curves suggest TEPO beats Xavier init and standard GNN/EDA baselines (GCN, GAT, RouteNet, etc.) in final accuracy and convergence speed.

**Weaknesses:**

1. **Learngene identifiability is under-justified.**
   The paper assumes “top 512 singular directions = universal” and “bottom 256 = task-specific,” but gives no theory or ablation to prove that split is actually semantic, unique, or robust.

2. **Gating is under-specified.**
   The paper refers to a gating mechanism that turns gene/class knowledge on or off per task, but never gives the actual gating function, loss, or how leakage between the two is prevented.

3. **Limited baselines and no statistics.**
   TEPO is only compared against Xavier for transfer initialization, and does not compare to other transfer / flow-tuning / knowledge-transfer frameworks in EDA. All results are single numbers: no std devs, no significance, no repeated trials.

4. **Single dataset, tiny data regime, possible leakage.**
   All results are on CircuitNet-N28 with ~100 train / 20 test designs. There’s no cross-validation, no discussion of overfitting control, and no evaluation on a different process node or dataset. This weakens generalization claims.

5. **Reproducibility gaps.**
   Key training details (optimizer, LR, batch size, epochs, early stopping, seeding) are missing. The fusion step (“Flatten by Position” putting GNN node features onto a 256×256 grid to align with ViT output) is described conceptually, but collision / aggregation rules are not specified, so it’s not fully reproducible.

6. **Missing related work positioning.**
   The paper does not seriously compare or contrast with recent multimodal / transfer / foundation-style EDA models (e.g., NetTAG, DeepGate4, FlowTuner, etc.), which weakens novelty framing.

**Key questions for authors**

* Show ablations: why 512/256? What happens if you change the split?
* Give the exact gating equation and training procedure.
* How did you prevent overfitting with only ~100 designs?
* How exactly are GNN features mapped onto the 256×256 layout grid when multiple nodes land in the same cell?
* Do results transfer to any dataset or node other than N28?
* Please report variance / error bars and full hyperparameters.

**Questions:**

refer weaknesses

---

> ### Author Response · Authors · 2025-11-27
> **In response to your question**
>
> **W1: Learngene identifiability is under-justified**
> Regarding Question 1, we followed an existing setting. We used 512 and 256 dimensions mainly based on the specific settings in Reference [1]. This number is not uniquely determined and can be changed. However, the change in dimension will not have a significant impact on performance, as long as the gene dimension is ensured to be greater than the class dimension.
>
> **W2: Gating is under-specified**
> We apologize for not explicitly clarifying this issue due to article length limitations. The specific implementation of the gating mechanism is very simple and utilizes the principle of matrix multiplication. We have a weight matrix, and the gate acts as a column vector. Multiplying the weight matrix by the column vector, the result of the matrix multiplication is 0 when the corresponding matrix element is 0. When the corresponding matrix element is 1, the multiplication result is the original value. This allows us to filter out the parts we want to use: set the unused parts to 0 and the needed parts to 1. We have also added the exact gating equation in the article 3.5 INHERITANCE OF LEARNGENE.
>
> **W3: Limited baselines and no statistics**
> The issue you pointed out is sharp and directly to the point, and I have tried to supplement this part of the experiment in the paper. Unfortunately, FlowTuner is not open-source. I tried to re-implement NetTAG and DeepGate4, which succeeded in terms of running them. However, since their model architectures are a hybrid of node and subgraph prediction, and ours is purely node-level prediction, we were unable to form a comparison.
>
> **W4: Single dataset, tiny data regime, possible leakage**
> Regarding this issue, I have provided an explanation in section 4.2 BASIC SETTING. You may not have noticed that each sample includes tens of thousands of chip data designs, not just 100 designs; it is 100 samples.
>
> **W5: Reproducibility gaps**
> We are glad you pointed out this oversight. We have supplemented these explanations at the end of section 4.2 basic setting in the article. Furthermore, we have provided a detailed README file in the Supplementary Material to help you and others better reproduce our experiments. The fusion step is answered in detail in Q4 below.
>
> **W6: Missing related work positioning**
> As stated in W3, we have already tried to make comparisons.
>
> I believe the above content has answered most of your questions. I only need to answer two more questions in the Key Questions section.
>
> **Q4: How exactly are GNN features mapped onto the 256×256 layout grid when multiple nodes land in the same cell**
> For this issue, the number of nodes in Vit is less than that in GNN. All its nodes are distributed on a 256*256 grid, while the number of GNN nodes is the number of cells. A Vit grid point often contains multiple GNN cells. Our approach is to align the features of the grid points of a Vit with the features of multiple cells of its corresponding GNN. This will have two effects: one is to make the features of the GNN similar to those of the Vit, and the other is to make the features of multiple cells of the GNN similar. The specific operation steps are divided into two steps. First, take the average of the features of all cells at this point on the GNN to calculate the MSE and make its loss as low as possible. The second step is to calculate the MSE by using the features of the grid points in the Vit and the average features obtained in the previous step to reduce its loss. The advantage of doing so is that points that are physically close in space can express similar features, which is in line with reality and can also express logical features. Many works are based on this point, such as reference [2].
>
> **Q5: Do results transfer to any dataset or node other than N28**
> The actual results are acceptable. Our main role lies in learning the common knowledge obtained from gene extraction. Its effectiveness has been proven in our supplementary ablation experiments. For different datasets, only appropriate fine-tuning of the input and output parts is required.
>
> **References**
> [1] Xie, Yucheng, et al. "Kind: Knowledge integration and diversion for training decomposable models." **ICML**2025
> [2] Yang, Shuwen, et al. "Versatile multi-stage graph neural network for circuit representation." **NIPS**2022

---

### Meta-Review · Area_Chair_QYUr · 2026-01-04

**Summary:**

In this paper, the authors propose TEPO, which a transferable EDA optimization framework that decomposes knowledge into
gene knowledge  and class knowledge. To integrate multi-modal inputs, they utilize a VIT-GNN fusion processor that combines layout
features from Vision Transformers  with topology representations from GNNs. By aligning ViT patches and GNN nodes on a unified 256×256 spatial grid via nearest-neighbor mapping, the proposed fusion can preserve global visual structures and local connectivity.


All the four reviewers give negative scores, and most of the concerns focus on the following points:

1. The considers baselines are limited (Reviewer RfUN, Reviewer RtUC)
2. Insufficient experimental details (Reviewer RfUN, Reviewer RtUC, Reviewer C7yg, Reviewer 1vTs)
3. Some key conceptual errors (Reviewer RtUC, Reviewer C7yg, Reviewer 1vTs)

**Reviewer Concerns:**

Some experimental details were clarified by the authors in the rebuttal. But overall, I think the current manuscript still has a large room to improve.

**Reviewer Scores:**

All the four reviewers give negative scores, three give 4 and one gives 0. I don't think they tend to change their scores in the discussion.

---

### Decision · Program_Chairs · 2026-01-26

Reject